# A Study of GDPR Compliance under the Transparency and Consent Framework

## ABSTRACT

This paper presents a study of GDPR compliance under the Interactive Advertising Bureau Europe's Transparency and Consent Framework (TCF). This framework provides digital advertising market participants a standard for sharing users' privacy consent choices. TCF is widely used across the Internet, and this paper presents its first thorough evaluation, investigating both the compliance of websites with TCF and its impact on user privacy. We reviewed 2,230 websites that use TCF and accepted the automatic decline of user consent by our data collection system. Unlike previous work on GDPR compliance, we found that most websites using TCF properly record the user's consent choice. However, we found that 72.8% of the websites that were TCF compliant claimed legitimate interest as a rationale for overriding the consent choice. While legitimate interest is legal under GDPR, previous studies have shown that most users disagreed with how it is being used to collect data. Additionally, analysis of cookies set to the browsers indicates that TCF may not fully protect user privacy even when websites are compliant. Our research provides regulators and publishers with a data collection and analysis system to monitor compliance, detect non-compliance, and examine questionable practices of circumventing user consent choices using legitimate interest.

## KEYWORDS

Privacy Regulation, GDPR Compliance, Consent Management Platforms, Transparency and Consent Framework, Ad Tech

**ACM Reference Format:**
Anonymous Author(s). 2023. A Study of GDPR Compliance under the Transparency and Consent Framework. In *Proceedings of the ACM Web Conference (WWW '24).* ACM, New York, NY, USA, 10 pages. https://doi.org/XXXXXXX.XXXXXXX

## 1 INTRODUCTION

In recent years, privacy regulations have been enacted by governments around the world to protect their citizens. The most well-known such regulation is the General Data Protection Regulation (GDPR), for protecting European Union (EU) citizens from unnecessary and unauthorized personal data collection [3]. GDPR requires *opt-in* consent from citizens of the EU before a company's website can collect or use users' personal data for various purposes including personalized advertising.

To address the GDPR requirements, when a user visits a website, publishers use consent management platforms (CMPs) from companies such as OneTrust, Quantcast, and Didomi to ask the user for permission to collect, store, use, and share their personal data, as illustrated in Figure 1. The user can approve, partially approve, or decline data collection and sharing. User consent is used to determine if personalised ads can be shown in programmatic ad auctions, including real-time bidding, as illustrated in Figure 2.

GDPR compliance requires a well-adopted solution to maintain and distribute user consent across different market stakeholders (e.g., publishers, ad tech companies, and advertisers) as a synchronized record, in order to ensure the user consent is honored. This is a complex real-time process requiring interoperability and consistency across different market stakeholders using different technologies. The technical aspects required to protect a user's privacy while supporting ad optimization are complex [24]. The lack of a common solution would create two challenges in GDPR compliance. The first is that communicating consent among many different parties is complicated and technical errors could lead to unintended non-compliance. The second is that a standard is required to audit the marketplace adoption of the solution.

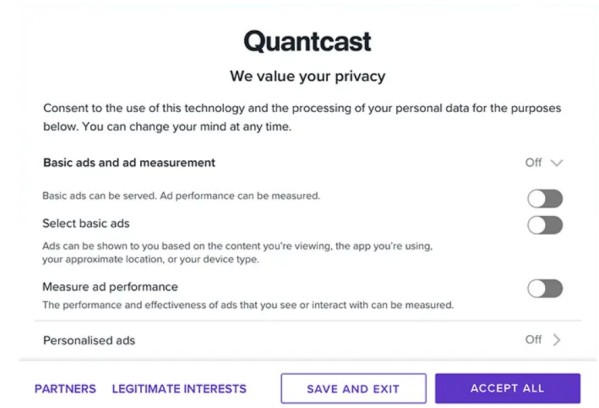

**Figure 1: Consent collection prompt example**

To address these challenges, the Interactive Advertising Bureau Europe (IAB EU) worked in partnership with the IAB Tech Lab [14], the main organization that develops industry standards for digital advertising across the world, to create the Transparency and Consent Framework (TCF) as a general and consistent GDPR consent solution. TCF enables publishers, ad tech companies (such as Magnite and Pubmatic), and advertisers to communicate the consent choices of EU citizens to other companies in the supply chain of online advertising and related activities [47]. TCF creates a standardized way for CMPs to capture user consent for different personal data collection purposes. Users' consent choices are encoded into a string of characters called the *TC string* [10, 16], which

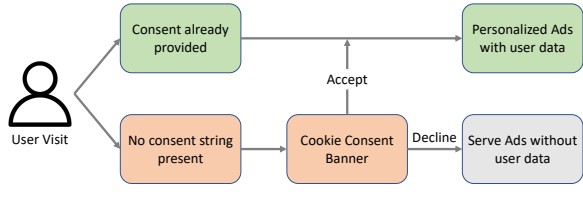

**Figure 2: Consent collection by a publisher**

is stored in the user's browser [16]. The stored TC string is used during the browser's HTTP communications with the website and the website's ad tech suppliers.

This paper evaluates if a large sample of publishers and their CMPs, using the commonly-used TCF 2.1 version, accurately record the decline of user consent to personal data collection. Understanding how websites use a consent framework such as TCF is an important part of ensuring user privacy on the Internet, as noncompliance with TCF or accessing loopholes in TCF may enable websites to violate user privacy as stated by GDPR. This study is further driven by two recent and complementary developments in the online advertising industry. First, TCF has become widespread as a solution for GDPR compliance [32]. Second, TCF has been challenged in court as not being fully compliant with GDPR because of the different views of the data processing authorities in Belgium and the IAB on whether the IAB is a data controller for TCF based on their respective interpretations of GDPR [8] and if the TC string is a user's personal data.

In this study, we conducted a simulation of an EU-based user interacting with CMP consent banners and declining consent. We created our dataset using servers within the EU to scrape thousands of global websites that collect and transmit user consent choices. We then monitored the TCF implementation of the websites for which it was evident that they had stored and communicated the user's consent election to other market stakeholders via a TC string. Specifically, we evaluate if publishers and their CMPs accurately record and communicate TC strings when the users decline consent to share their personal data.

The high-level insight from our results is that most TCF sites are legally compliant, but a large majority use legitimate interest, allowed under GDPR, to collect user data for commercial interests (not for functional/performance reasons), despite the users denying consent. This is not in the spirit of GDPR, and previous studies have shown that users feel "cheated" by such practices [36]. Additionally, tracking cookies are set to the user's browser even when visiting TCF compliant websites. Thus, there is no guarantee that user privacy is protected by TCF.

Specifically, we found that 97.8% of the websites studied complied with TCF v2.1 in that they properly stored and communicated the user consent choice. However, we found that 72.8% of the TCF compliant websites claimed legitimate interest as a rationale for overriding the consent choice. This compromises user privacy by allowing the processing and collection of their personal data without their consent. The use of legitimate interest is a legal consideration regarding GDPR compliance and is not a technical matter that affects TCF compliance. Analysis of the cookies setting practices when visiting each website in our sample showed that even

when visiting TCF compliant websites, an average of 1.09 tracking cookies are set, which implicitly compromises user privacy. We also observed several direct non-compliance cases, including one high-profile website where user consent choice was not honored.

We further found that the IAB decoder tool for TC strings sometimes decodes invalid TC strings, which needs to be addressed by IAB to ensure a user's consent choices are fully honored.

To the best of our knowledge, this is the first study of TCF compliance and how it relates to GDPR compliance since the updates to the TCF brought about by versions 2.0 and 2.1. We believe our research can help regulators (e.g., European Data Protection Authorities) and market participants make informed decisions about using TCF for complying with GDPR, and by extension the IAB Tech Lab's Global Privacy Platform (GPP) [11], a superset of TCF. Furthermore, our data collection and analysis system[1] will be open-sourced to allow researchers, regulators and market participants to monitor compliance with GDPR.

## 2 BACKGROUND AND RELATED WORK

### 2.1 GDPR

Under GDPR [3, 4], user consent is required for processing or collecting personal data for advertising purposes. There are 12 purposes (including two special purposes) for personal data collection, as shown in Table 1. A user can consent to data collection for each purpose individually. Not all personal data collection purposes require user consent. A legal basis for personal data collection that does not require user consent is called legitimate interest. A website can claim legitimate interest anytime its interests outweigh those of the data subjects for situations such as direct marketing and information technology security [2, 3]. This includes commercial or government operated websites [48]. A website is not required to ask for consent for the two special purposes of data processing as their legal basis is always provided by legitimate interest and the user is not able to opt-out of data processing activities covered by these two purposes [28].

GDPR regulators fine companies for non-compliance. For example, enforcement of GDPR resulted in CNIL (i.e., the French privacy regulatory authority) fining Google 150 million euros for making it more difficult to reject than accept consent [7]. Such enforcement of GDPR motivates websites to adopt TCF to aid with compliance.

### 2.2 TCF

TCF is a cross-industry voluntary standard that is intended to enable publishers of websites and apps (first parties) and technology partners that support the delivery, personalization, or measurement of advertising and content (third parties or vendors) to work together and provide users with a standardised experience when they make privacy choices [14]. In short, it is an open-standard technical framework that enables websites, ad tech, advertisers, and ad agencies to obtain, record, and update consumer consent for personal data use.

There have been several versions of TCF. TCF v2.0 allowed users to gain more control over which vendors (e.g., ad tech companies) could process or collect data for each purpose [12, 16]. TCF v2.1

---

[1]Github URL is removed now for the double-blind requirement.

is an improvement over TCF v2.0 that addresses a 2019 ruling by the European Court of Justice [1, 14, 16, 40]. In TCF v2.1, CMPs make additional disclosures to the user. One of them is when alternatives to cookies, such as the user's local storage within their browser, are used to collect and share personal data. This paper investigates TCF v2.1, which is widely used across the Internet. To the best of our knowledge, this is the first paper to investigate TCF compliance since the improvements of TCF v2.0 and v2.1. While TCF v2.2 has become available in May 2023 [16], websites are not required to implement it until November 2023 [15]. Our dataset was collected before the implementation of TCF v2.2. Given that each version of the TCF builds on the previous versions, our findings are still relevant. Additionally, our method of collecting and analyzing the datatset to evaluate compliance will still work for evaluating compliance under TCF v2.2.

Our research occurred during a critical period for TCF. In February 2022, the Belgian Data Protection Authority (DPA) claimed that TCF v2.0 violated GDPR and required IAB EU to submit an action plan for remedy [8]. In September 2022, the Belgian appeals court deferred to the Court of Justice of the EU regarding two specific questions [6]. One relates to data controller status for the IAB EU, and the other to whether the TC string represents personal data under GDPR. In January 2023, the Belgian DPA announced that it approved the IAB EU's TCF action plan, but this action plan is now temporarily suspended because the IAB EU made an appeal to the Belgian Market Court. In September 2023, the Belgian Market Court suspended its review pending responses from the Court of Justice of the European Union (CJEU) [9, 13, 27].

With the fate of TCF uncertain, our research is timely and may help shape TCF's future. While the court cases are not directly related to our study, providing a large-scale analysis of TCF compliance can help legal experts with context and a deeper understanding of TCF adoption. Our research also contributes to a better understanding of the technical foundation of TCF for GDPR compliance.

## 2.3 Related Work

The most related work to ours [40] measured the compliance of websites that use CMPs registered with the IAB EU. The authors found 1,426 websites using consent banners from CMPs registered with the IAB EU. Out of this set, 10% of websites stored the TC string before the consent choice, 5% did not create a TC string that accurately reflected the user's choice, 7% did not offer a way to opt-out of data collection, and almost 50% had pre-selected choices. Our paper differs from this research by performing a study with the updated TCF after the improvements of v2.0/v2.1. More importantly, our results show higher rates of compliance and we investigate how frequently the use of legitimate interest allows data processing without user consent. One additional paper investigated TCF adoption [32], but unlike our work, it did not address compliance [32].

There are also works that study the relationship between TCF and GDPR. Ryan and Santos [47] argue that TCF cannot be monitored, secured, nor audited, while other works state that TCF for real-time bidding is unlikely to be capable of becoming GDPR compliant without massive changes [22, 47, 54, 55]. Santos et al. [51] explore the different roles of CMPs, as defined by the TCF and GDPR. Under TCF, vendors and publishers commonly cite advertising as the reason for their legitimate interest, but Kyi et al. claim

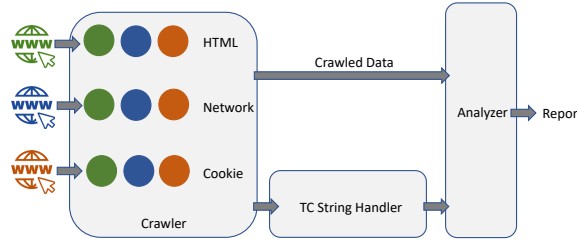

**Figure 3: System Design**

that this is not compliant with GDPR [36]. Our work is different from these studies as we focus on TCF compliance by publishers and their CMPs.

Several papers check GDPR compliance using collected information related to consent banners [18, 35, 39, 43, 50] or collected information on cookies and tracking [19, 20, 23, 45, 49]. Also, ad blockers in web browsers are often utilized to prevent tracking, and research exists that studies countermeasures taken by advertisers and publishers [38]. Instead, our study evaluates TCF compliance by analyzing if users' consent choices in TC strings are honored. There are also many studies exploring aspects of GDPR compliance that are not related to TCF [39]. These include topics such as familiarity of website managers with GDPR [30, 53], deceptive practices used to nudge users toward consenting to personal data collection [31, 52] due to inadequate awareness concerning data privacy [46] vaguely worded regulations [17] within GDPR, and solutions to GDPR compliance [29, 56]. In addition, research has been published to assess the design of web browser privacy protections that can also support the needs of the advertising industry [34]. Our paper addresses the different and timely research topic of TCF compliance, which is related to GDPR compliance.

## 3 SYSTEM DESIGN

Our system automatically acquire data from publishers and their CMPs using TCF to determine if a user's decline of consent is properly recorded, which is required by GDPR. The technical means through which a user's consent information is communicated in TCF relies on common web technologies, such as cookies and HTTP requests. When a user clicks a consent banner, the consent information is stored in a cookie in the user's browser. The details of the user's consent are encoded in a special format to facilitate storage and transmission. The encoded value is called the *TC string*.

TC strings are used by marketplace participants to know whether a user has provided consent. The data collected by our system can monitor a TC string's lifecycle: when it was created, to whom it was communicated, and when it was sent. The system observes which ad tech vendors provide appropriate HTTP responses acknowledging the TC string and properly forwarding it to a third party, such as another ad tech vendor. TCF offers standardized guidance to market participants for the communication of TC strings.

Collecting data for a large number of websites is challenging due to variations in the TCF naming conventions of the fields (e.g., parameters in the query strings, cookie names) and variations in consent banners (different user interfaces and selection options).

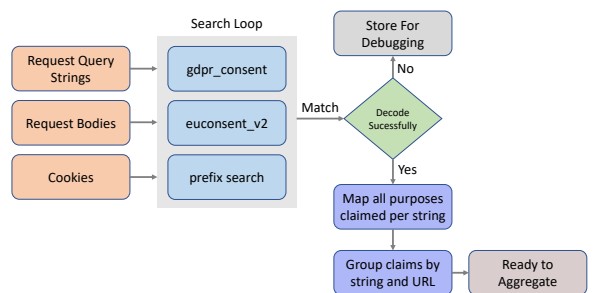

Figure 4: TC String Handler Workflow

To address this problem, we use a headless browser [21, 41], functioning as a web client, to record and analyze all the HTTP requests that occur when a publisher's website loads, which includes request and response headers, bodies, and metadata (such as which script or action initiates an additional request). As third-party ad tech vendors participating in the advertising supply chain of a publisher's website look for and communicate the TC string, our system inspects the HTTP request data to determine if the TC string is present and, if it is, that it has not been altered. There also exist ad tech vendors, called "fourth parties", participating in the publisher's advertising supply chain indirectly, through the interaction with the third-party ad tech vendors with direct access to a publisher's ad inventory. Our system is not able to detect fourth parties. When a TC string is not correctly recorded, it has cascading consequences with ad tech companies that are direct and indirect participants in a publisher's ad auction.

Figure 3 shows the flow diagram of our system, which has three major components: (i) a crawler that visits websites and collects data from the browsers, (ii) a TC string handler that parses the crawled data to identify the TC strings, decodes the TC strings, and stores the decoded data in a database, and (iii) an analyzer that aggregates the data to generate reports.

## 3.1 Crawler

Under GDPR, websites utilize a CMP to display the consent banners to users presenting options to accept or deny consent. Each CMP offers specific HTML components within the banner. Users have the right to specifically select the types of data uses they consent.

To streamline the process of collecting large amounts of data, we developed a crawler that automatically loads target websites and simulates the user's decline of consent. To implement the crawler, we use TypeScript Playwright [41], which allows developers to write browser automation scripts. This crawler identifies various types of consent banners, applies specific rules to simulate the decline actions, and stores user data requests and cookie information for further analysis.

The crawler utilizes SOCKS5 proxies to load the target websites from servers located in the EU. To recognize consent banners, the crawler examines the request URLs when a website is being loaded and identifies the URLs associated with different consent banners. Since each CMP supports one or more templates, the crawler verifies the presence of HTML IDs assigned to visible user interface components to determine the specific variant of the loaded consent

CLcVDxRMWfGmWAVAHCENAXCkAKDAADnAABRgA5mdfCKZuYJez-
NQm0TBMYA4oCAAGQYIAAAAAAEAIAEgAA.argAC0gAAAAAAAAAAA.
IFukWSQgAIQwgI0QEByFAAAAeIAACAIgSAAQAIAgEQACEABAAgA
QFAEAIAAAGBAAgAAAAQAIFAAMCQAAgaAQiRAEQAAAAANAAIAAggA
IYQFAAARmggBC3ZCYzU2yIA

Figure 5: TC string highlighting the core vendor and consent details

banner. A set of rules are defined for templates in major CMPs, which are used by the crawler to interact with the consent banner with a series of actions, such as clicking on a specific button, toggling a switch, or selecting a checkbox, in order to decline the consent. Finally, the crawler verifies the successful execution of the actions and stores the name of the CMP, the status of the consent action, as well as the user data requests and cookie information.

## 3.2 TC String Handler

To automate the extraction of TC strings from the crawled data and prepare it for analysis, our system follows the pipeline shown in Figure 4. First, it parses the data collected by the crawler and extracts the TC strings from both the cookies and HTTP requests. Specifically, it scans for all URL requests and cookies to get the values that correspond with the key "gdpr_consent" or "euconsent_v2". Then, the parser analyzes the values and identifies the strings that begin with the prefix "C", which serves as the initial character of the TC string with TCF version 2.0 or higher.

Every TC string consists of three segments separated by a "dot" character, as shown in Figure 5. The first segment is called the core string and contains the core vendor transparency and consent details. The last two segments, 'Publisher purposes transparency and consent' and 'Disclosed vendors' are optional.

The TC strings are decoded using IAB's TC string decoder [33] to get the consent-related information in a standardized format. A decoded TC string contains a map corresponding to a list of "purposes" for collecting consent. Table 1 shows the 10 purposes [25, 28] that can be selected, along with two special purposes. Each of these ten regular purposes requires a legal basis, which can be either consent or legitimate interest. The decoded data is then stored in a key-value database, where each row represents a TC string, and the key-value pairs correspond to the column names and their respective values. To determine how user consent is recorded and transmitted to other market participants, our system inspects the core string and stores the values for the consent categories (listed in Table 1) in separate columns. Additionally, a column is included to store the name of the website that generates the TC string.

## 3.3 Analyzer

The analyzer utilizes a series of Python scripts to analyze the data collected at different stages. First, it assesses the performance of the crawler and provides a report on the count of websites where a consent banner is identified and then consent is successfully declined. Next, the analyzer looks for non-compliance in the relaying of TC strings between the CMP platforms and ad tech vendors within a single web session. Our system achieves this by analyzing the decoded TC strings and checking whether user consent or

| Purpose | Description |
|---------|-------------|
| Purpose 1 | Store and/or access information on a device |
| Purpose 2 | Select Basic Ads |
| Purpose 3 | Create a personalized ads profile |
| Purpose 4 | Select personalized ads |
| Purpose 5 | Create a personalized content profile |
| Purpose 6 | Select personalized content |
| Purpose 7 | Measure ad performance |
| Purpose 8 | Measure content performance |
| Purpose 9 | Apply market research to generate audience insights |
| Purpose 10 | Develop and improve products |
| Special Purpose 1 | Ensure security, prevent fraud, and debug |
| Special Purpose 2 | Technically deliver ads or content |

**Table 1: Consent purposes in TCF**

legitimate interests for data collection and processing are claimed. Furthermore, the analyzer aggregates the decoded TC strings if there are different TC strings resulting from one crawl. The result of aggregation is a TC string that claims any purpose that is claimed by at least one of the TC strings from the crawl. We do this because from the user's perspective, even a single company processing their personal data means their privacy is compromised.

## 4 FINDINGS

### 4.1 Data

Websites were selected for our dataset using the Tranco list [37], which provides a ranking based on traffic volume of popular worldwide websites and is specifically designed for research purposes. It has been used in this research area before [20, 36, 40]. Then, we selected the subset of websites on the Tranco list that are actively transacting in the open programmatic marketplace using aggregated ad inventory availability data from DeepSee.io, an online publisher intelligence company. Starting with this list, we selected all websites meeting our criteria for analysis. Since we were automating the process of an EU user not consenting to data collection and processing (referred to as "declining consent"), we needed to be able to automate interacting with the consent banner. To simplify this time consuming process, we focused our efforts on banner variations from the CMPs which were found most often. Each CMP also offers many variations on the types of consent banners to their clients. Therefore, we only analyzed compliance on websites where we could successfully reject the specific consent banner shown. Additionally, we cannot analyze any websites that do not share the TC string with other stakeholders in the online advertising ecosystem. Given that these websites work with an IAB-registered CMP, it is possible that these websites employed TCF as their consent management solution, but we are unable to check if they are compliant in storing and transmitting the user's consent election because we could not see the string in cookies or the HTTP requests. This process resulted in the 2,230 websites that we analyzed for this paper.

Our crawler visited a URL associated with each of the 2,230 websites in the analysis. By design, it visited multiple URLs (or

web pages) on some of these websites. This resulted in a total of 8,929 crawls to web pages on 2,230 websites. Visiting multiple web pages per site allows to more accurately capture what will happen when a user visits a website. We do find different TC strings may be created on different webpages from the same website. For the rest of the analysis, we will refer to the analysis of the 2,230 websites as domain-level analysis and the analysis of the 8,929 web pages as crawl-level analysis.

### 4.2 Checking TCF Compliance

To evaluate the TCF compliance of the crawls, we needed to first decode the TC strings. During this step, we found that the IAB decoder sometimes provides a decoding of invalid TC strings. As illustrated in Figure 4, we found TC strings by searching the http requests and cookies for specific keys and prefixes, and then we confirmed the validity of the TC strings by testing if the IAB decoder would decode the string or give an error. Some of the TC strings we found were only fragments of TC strings that were stored as the values for the "gdpr_consent" or "euconsent_v2" keys. These TC string fragments should not be decoded by the IAB decoder, but we found that they were. Through manual inspection, we found 62 decoded strings that were not valid TC strings in TCF v2.1. They were easily identified as they either did not start with the letter "C" or were clearly not TC strings such as the text "cookie_banner_accepted". Although these strings were not valid TC strings, and thus could not be communicating user consent elections, the decoder wrongly stated that consent had been given for several data processing purposes. Currently, the IAB decoder lacks a solution for checking if a TC string is valid or not. This issue should be further evaluated by the IAB to fix the potential problems with the IAB TC String Decoder. We removed the 62 invalid strings from our data before starting our analysis. Since these strings were invalid, we did not count crawls or domains with only invalid TC strings towards the total number of crawls (8,929) or domains (2,230).

Table 2 shows an overview of our data and provides a breakdown by the different CMPs. All numbers outside of parentheses are for crawl-level analysis. Numbers inside of the parentheses are for domain-level analysis. The second column, "Crawls with Valid TC String" gives a count of how many of our valid 8,929 crawls (or 2,230 domains) were using each of the five CMPs. Each crawl can either have an empty or non-empty TC string. The number of crawls resulting in a non-empty TC string is stored in the "Crawls with Empty TC String" column. In total there are 693 crawls to 605 distinct domains that result in an empty TC string. All empty TC strings are considered TCF compliant because they do not claim the user consented to data collection and processing.

The crawls that resulted in a non-empty TC string can then be further split into crawls that were TCF compliant and crawls that were not. Citing legitimate interest as a legal basis to collect or process personal data is compliant with TCF, as long as it is not for Purpose 1. The only appropriate legal basis for data collection and processing for Purpose 1 is user consent [5, 26]. Thus, citing legitimate interest for Purpose 1 is an example of non-compliance. Additionally, in this dataset, TCF non-compliance is observed anytime a TC string states user consent was provided because we simulated a user declining consent. The number of crawls resulting in a TC

| CMP | Crawls with Valid TC String (Domains) | Crawls with Empty TC String (Domains) | Crawls with LI Claims (Domains) | Crawls Claiming Consent (Domains) | Crawls Claiming LI for Purpose 1 (Domains) | Percentage of Crawls (Domains) with Violation |
|---|---|---|---|---|---|---|
| Didomi | 8571 (1933) | 643 (555) | 7895 (1374) | 0 | 33(16) | 0.4%(0.8%) |
| CookieBot | 233 (224) | 32 (32) | 201 (192) | 0 | 0 | 0% |
| Quantcast | 14 (14) | 1 (1) | 10 (10) | 3 (3) | 0 | 27.3% (27.3%) |
| OneTrust | 30 (30) | 17 (17) | 13 (13) | 0 | 0 | 0% |
| Ringier Axel Springer Polska | 81 (29) | 0 | 0 | 81 (29) | 0 | 100% (100%) |
| **Total** | **8929 (2230)** | **693 (605)** | **8119 (1589)** | **84 (32)** | **33 (16)** | **1.3% (2.2%)** |

Table 2: Aggregate TC String Categories By CMP (Note: LI = Legitimate Interest)

string which only claimed legitimate interest for some subset of the purposes 2-10 is in the "Crawls with LI Claims" column. Any crawls where the TC string also claimed legitimate interest for Purpose 1 or that the user gave consent are not stored in this column. Thus, the column represents the total number of crawls (or domains) where a non-empty TC string is generated, but no violation occurs. If we only consider the 2,182 domains where no violation occurs, then we can see that 1,589 (72.8%) of those domains claim legitimate interest. This allows them to process personal data without user consent. It should be noted that there are 12 domains in our dataset that have at least one crawl where the result is an empty TC string and one where the result is a non-empty TC string that does not violate TCF policies. We count these domains as belonging to both categories for the domain-level analysis.

Because we simulated a user rejecting consent, a violation occurs whenever legitimate interest is claimed for Purpose 1 or when the TC string states user consent was given. The percentage of crawls and domains resulting in a violation of TCF policy are tracked in the "Percentage of Crawls (Domains) with Violation" column. The crawls and domains where consent is claimed are counted in the "Crawls Claiming Consent" column. All 81 crawls to the 29 distinct domains using the Ringier Axel Springer Polska CMP do not properly represent the user's consent election. While three out of the fourteen domains (27.4%) using the Quantcast CMP also result in similar violations, this is slightly less concerning as it is not every domain using the CMP and the sample size is relatively smaller. The crawls and domains where legitimate interest is claimed for Purpose 1 are stored in the "Crawls Claiming LI for Purpose 1" column. All instances of this are found at domains who use the Didomi CMP. However, due to the large number of domains using Didomi in our dataset, these violations only account for 0.4% of crawls to domains using Didomi and only 0.8% of such domains. Nevertheless, the problem is still important. Any visit to a website where legitimate interest is claimed for Purpose 1 means that data is stored on, or accessed from the user's machine for the purpose of identifying the user without their consent. In total there are 117 crawls from 48 distinct domains where TCF policies are violated, which is 1.3% of all crawls and 2.2% of all domains, respectively. This demonstrates that most websites are TCF-compliant.

## 4.3 Analysis of Premium Domains

Next, we decided to analyze if the non-compliant websites are major or minor websites. To do this analysis, we separated out a group of domains that we call "premium domains". These are domains ranking in the top 5,000 of the Tranco list. We then analyzed all

crawls visiting any web page on the premium domains. In doing so we could show that our findings are not exclusive to domains with little traffic. We determined that 2,315 crawls to 48 distinct domains qualified for this analysis. The analysis of these TC strings is summarized in Table 3, which is interpreted the same way as Table 2. The "Crawls Claiming Consent" column shows that 2 of the 84 domains recording user consent in the TC string when consent was not granted are premium domains. Thus, TCF non-compliance is not only occurring on small domains with not much traffic. However, there are no premium domains where legitimate interest is claimed for Purpose 1. The "Crawls Claiming Legitimate Interest" column shows that legitimate interest is claimed for at least one of purposes 2-10 by 25 premium domains without a TCF violation. This means only 25 out of 46 (54.3%) premium domains without a violation use legitimate interest to process users' data without consent. This is less than the 72.8% of all domains without a violation that use legitimate interest. Similarly, 26 out of 48 (54.2%) premium domains have at least one crawl resulting in an empty TC string, while this occurs in only 605 out of 2,230 (27.1%) of all domains. Although legitimate interest is claimed less frequently for premium domains, it still occurs in more than half of the premium domains that are TCF compliant. Thus, we conclude that the issues of TCF non-compliance and processing of personal data using the legal basis of legitimate interest, and not user consent, applies to high traffic domains.

| Crawls with valid TC string | Crawls with Empty TC | Crawls Claiming Consent | Crawls Claiming Legitimate Interest |
|---|---|---|---|
| 2315 (48) | 49 (26) | 36 (2) | 2230 (25) |

Table 3: Premium Publisher Analysis (Note: Numbers listed in parentheses are for domains)

## 4.4 What is legitimate interest used for?

Thus far, all legitimate interest claims for purposes 2-10 have been treated equally. This is because, under TCF v2.1, legitimate interest was a valid legal basis for all purposes other than Purpose 1 [26]. Nevertheless, analyzing how websites can legally process user data without consent is still important. For example, a website claiming legitimate interest for Purposes 3 and 4 can gather information about the user (e.g., what articles, videos, or products that the user views) to help infer user interests. The inferred interests can then be used to target the user with specific advertisements. Under TCF v2.1, this can all be done without user consent [26]. However, under the new TCF v2.2, legitimate interest will no longer be a valid legal basis for data processing purposes 3-6 [16, 28]. This change gives

further reason to explore the data processing purposes for which websites claim to have legitimate interest. Measuring the frequency of legitimate interest claims for purposes 3-6 under TCF v2.1 gives an idea of how significant the changes of the new TCF v2.2 will be.

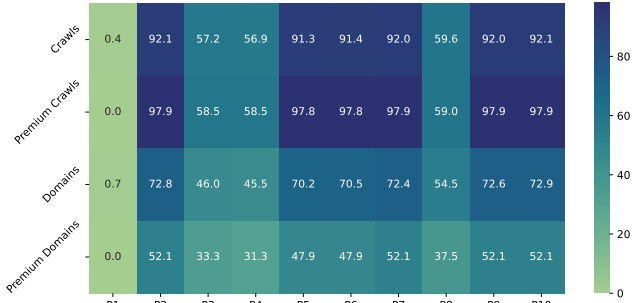

**Figure 6: Legitimate Interest Claims by Data Processing Purposes (Each number represents a percentage)**

Figure 6 displays the percentage of the resulting TC strings where legitimate interest is used as a legal basis for each of the data processing purposes defined in Table 1. The first row is for the percentage of resulting TC strings from all 8,929 crawls that claim legitimate interest for each of the 10 purposes. The second row conveys the same information, but only for the 2,315 premium crawls. Likewise, the third row is for all of the 2,230 distinct domains and the fourth row is for the 48 premium domains.

We observe that among all domains, legitimate interest is claimed as a legal basis for data processing purposes 1, 3, 4 and 8 at a much lower rate than for other purposes. Legitimate interest claimed for Purpose 1 is only 0.4% of crawls and 0.7% of domains. This is a good sign of general TCF compliance in the domains studied. Data processing under purposes 3 and 4 enables serving personalized advertising content based on a user profile. A plausible explanation for their relatively low percentage of legitimate interest claims is that users are unhappy if they discover that profile information was collected without their consent. It is unclear why there is a lower percentage of crawls where legitimate interest is claimed for purpose 8. We also observed that, in general, premium domains are less likely to claim legitimate interest than other domains. This may be because premium domains feel they have a higher reputation to uphold, and thus are less willing to process or collect user's personal data without consent.

A final conclusion from this analysis is that many domains, including the premium domains, will have to change their current practices under TCF v2.2. Since legitimate interest is no longer a valid legal basis for data processing purposes 3-6, over 70% of the domains in our sample who used this legal basis will need to change their practices. It is likely that the rate of non-compliance under TCF v2.2 will be higher, at least in the beginning, because websites are used to claiming legitimate interest as a legal basis to create a user profile for selecting targeted advertisements and content.

### 4.5 What do our results mean for user privacy?

Our results are also interpretable through the lens of how they affect user privacy. For example, the heavy reliance on legitimate interest as a legal basis (subsection 4.4) is concerning because it allows some processing of personal data without user consent. For example, websites may combine user information obtained offline (e.g. inferences about the user's interests from data vendors) with the information collected when the user visits their website to select advertisements under Purpose 2 with Feature 1 [28]. Some users may feel this is a violation of their privacy, and this activity is still allowable without user consent in the new TCF v2.2.

While we cannot tell whether user privacy is compromised each time a website or ad tech vendor claims legitimate interest, there is a way to check if the recording and communicating of the user's consent choice is related to user privacy. The crawler collects information from the browser about which cookies are set. Such cookie information is used to show that user privacy is not perfectly protected even when no violations occur. Not all cookies affect user privacy, though. Some cookies are required for the website to work properly (e.g., "euconsent_v2" cookie storing the TC string value).

We develop a method of classifying cookies based on whether they negatively affect user privacy or not. We used a database called Cookiepedia [44] for this purpose, as it has been used in similar research for identifying tracking cookies [42]. Cookiepedia follows the classification standards set forth by the UK International Chamber of Commerce. Thus, all cookies are classified as strictly necessary, functionality, performance, or targeting/advertising. The strictly necessary cookies are required to provide basic services of the website. Functionality cookies improve the user's experience (e.g., enabling the website to be presented in the user's preferred language each time it loads) [44]. Similar to an existing paper using Cookiepedia [42], we do not consider cookies classified into these two categories to be endangering user's privacy. Performance cookies are only used in the aggregate for improving performance aspects of the website [44]. Since such data can be anonymized, we do not consider performance cookies to threaten user privacy. The cookies labeled as "targeting/advertising" by Cookiepedia are the only class of cookies that we consider to inhibit user privacy. We refer to them as "tracking cookies" for the rest of the paper. There are also many cookies which Cookiepedia classifies as unknown because it relies on self-reported descriptions of the cookies that are not always available. Thus, any estimates on the number of tracking cookies is conservative.

Now, we analyze domains based on the average number of tracking cookies set to browsers across all the crawls to the domain. There are three categories of domains. 1) domains where empty TC strings are generated, 2) domains where non-empty TC strings are generated that do not violate any TCF policies as described in Section 4.2, and 3) domains where the generated TC strings violate some TCF policy as identified in Section 4.2. The results for these domains are shown in Table 4 .

In the first category of domains, the generated TC strings are empty, and thus, no legal basis is claimed for data processing under any of the ten purposes. This first category corresponds to row of Table 4 titled "Empty TC". We found that there were 0.58 tracking cookies set on average for domains in this category. Tracking cookies were only set in 30.8% of the domains belonging to this category. This raises the question as to why there would be any tracking cookies set when the website and their associated ad tech vendors do not claim any legal bases for data processing. It seems unlikely

| Category | Number of Domains | Average Number of Tracking Cookies | Percentage of Domains with Tracking Cookies |
|---|---|---|---|
| Empty TC | 605 | 0.58 | 30.8% |
| Non-empty, no violation | 1589 | 1.28 | 59.6% |
| Violation | 48 | 2.39 | 83.0% |
| **Total** | **2230** | **1.11** | **52.3%** |

**Table 4: Tracking Cookies**

tracking cookies would be set for special purposes of data processing that are always available to publishers and ad tech vendors (and therefore are not part of the TC string). Those special purposes only ensure security and technically deliver content and advertisements [28]. It is possible that the tracking cookies are set by bad actors, who set their cookies regardless of what the TC string says.

The second category of domains are those that generate a non-empty TC string that does not violate any TCF policies. This means that legitimate interest is claimed for at least one of the data processing purposes 2-10. This category corresponds to the row titled "Non-empty, no violation" in Table 4. We found that tracking cookies were set in 59.6% of the domains belonging to this category. On average 1.28 tracking cookies were set. Both numbers represent a significant increases over the ones in the first category. This implies that some publishers or ad tech vendors believe that legitimate interest is a valid legal basis for setting tracking cookies. This does not align with our interpretation of the TCF data processing purposes. A legal basis for Purpose 1 must be given to store cookies that can be used to identify a user's device each time they visit a website. Therefore, it is questionable why claiming legitimate interest would ever allow for setting tracking cookies.

The final category of domains is those that violated TCF policy in how they stored and communicated the user's consent choice. This category corresponds to the row titled "Violation" in Table 4. Since there is a TCF violation in all of these domains, we expected a higher number of tracking cookies to be set. This is because the improper TC string (which claims a legal basis for Purpose 1) states that setting tracking cookies is allowed. It is important to note that whoever sets the tracking cookie could be responsible for altering the TC string to claim a legal basis for Purpose 1. In such cases it is not the fault of the domain (nor their CMP) in our sample, but is the fault of some third party actor that they work with. Our expectations were confirmed in the analysis, as we found that an average of 2.39 tracking cookies were set when visiting domains where TCF violations were found. Such cookies were set by 83.0% of domains in this category. Although this result was expected, it shows the importance of domains and their CMPs properly recording the user's consent election. When the string is improperly recorded after a user rejects consent, more tracking cookies are set and user privacy is further impacted.

The analysis of tracking cookies set to the user's browser leads to some questions about the effectiveness of the TCF. It is concerning that there are tracking cookies being set after the user declines to consent to all of the TCF's purposes of data processing. It is not surprising to see tracking cookies set when TCF policy is violated, and this is not as concerning because we found TCF policies were

violated in only 2.2% of the studied domains. What was surprising was that tracking cookies were set even when no TCF policies were violated. The average number of tracking cookies set when no violation occurs is the average number of cookies set in domains belonging to the first two rows of Table 4. This is equal to 1.09 tracking cookies. Finding a significant number of tracking cookies set when visiting domains that follow TCF policy shows that there are either bad actors in the industry or there are loopholes in the TCF.

This calls for future research on several questions. One question would be to determine if tracking cookies are set to compliant domains by a particular few companies. Another interesting research question for legal scholars is to determine if significant loopholes exist in the TCF policy that allows certain tracking cookies to be set without consent. Regardless of the outcomes, the IAB EU must address such issues to protect user privacy.

## 5 CONCLUSION

We conducted a study to evaluate TCF as a consent-sharing standard for GDPR compliance. Our study showed a high rate of TCF compliance by publishers and their CMPs: 2.2% of the websites in our sample did not comply with users' declining consent choice. However, 72.8% of the websites where no violations were found circumnavigated the user choice by claiming legitimate interest for at least one data processing purpose. Given this frequent use, legitimate interest claims merit further scrutiny by regulators. The newest version of the TCF, TCF v2.2, addresses part of this issue as legitimate interest is no longer a valid legal basis for purposes 3-6. Given that we found legitimate interest claims for these data processing purposes in over 50% of the crawls, an interesting topic of future research would be to analyze if websites become compliant with this new policy and determine what impact this has on advertising revenue.

Similarly, we found that despite high rates of compliance with TCF, there were many instances of cookies being set that may compromise user privacy. This is an issue for users who are concerned with their privacy and trust that GDPR and the associated consent frameworks will protect their privacy. We identified this finding as one that the IAB EU should address.

We plan to make the software and the collected dataset publicly available, which will help marketplace participants better assess their compliance with GDPR, help regulators in their efforts to monitor TCF adoption and compliance by websites, and inspire future research. Our study already has practical implications because we found that the IAB decoder sometimes provides a decoding of invalid TC strings, which needs to be addressed by the IAB to ensure users' consent choices are being honored. As the European courts are currently evaluating TCF's compliance with GDPR, we believe our study and system will provide useful insights for regulators and for the industry at a critical time for addressing user privacy concerns in the digital publishing and advertising industries.

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
