# OpenReview forum: "A Study of GDPR Compliance under the Transparency and Consent Framework"
_ACM.org/TheWebConf/2024/Conference — TheWebConf24 Oral_

### Official Review · Reviewer_qpZP · 2023-11-20

**Novelty:** 5
**Technical Quality:** 7

**Review:**

“A Study of GDPR Compliance” presents a study on TCS compliance including an assessment of the uptake from a significant sample and its actual implementation in terms of cookies. The goal of the study is to provide a picture of the current state of the standard, with a particular focus on the override of preferences claiming legitimate interest under the GDPR clause.

The paper presents a well-structured method for a systematic study of TCF implementation and (correctness and compliance). The study looks into how the user choice about cookies is stored in TC strings and provides findings about the justifications for avoiding user explicit rejection based on a range of clauses of legitimate interest.

The topic is on the spot with the conference and the track. My only comments concern its limitations and novelty. On the one hand, the study follows the path of previous works and confirms their findings. On the other hand, the study is limited to how declarations are handled (at least in my understanding), avoiding looking at other related practices such as the transmission of user data back to servers. In a nutshell, my main question is about what is actually being assessed by this work: what is the relation between TCF and actual practice (beyond cookies)? Are there ways to avoid the system while keeping compliance with TC strings? I am left with these doubts and it is hard to guess the actual scope of the contribution (which is there at least on the technical solution but hard to quantify concerning the more broad discussion about privacy).

Overall, the technical quality of the paper is appropriate to the level of the conference. The method used to collect data is very well described and documented. The writing, figures and tables are good and the structure is well thought out.

**Questions:**

-	What is the novelty of the paper? The study confirms previous findings about publishers' bad practices of ignoring explicit rejections. In general, it is hard to identify the novelty of the work as the paper (very well) reports about similar studies. The technical work on TC strings adds little to the findings.
-	Are there any recommendations for browsers? It seems to me that a clear finding of this and other works is that self-regulation is not fit for purpose. Are there any reasons why this issue cannot be addressed by browser companies with one consistent and verified approach? Should not this be the role of a regulator as the EP?
-	What is the actual scope of this work in terms of uncovering practices? It seems from its introduction that TC strings collect declarations. However, did the authors look into actual traffic back to servers? Please clarify this point in case it is relevant or not and why.

**Ethics Review Description:**

No issues, data are being generated by researchers activities

**Reviewer Confidence:**

3: The reviewer is confident but not certain that the evaluation is correct

**Scope:**

4: The work is relevant to the Web and to the track, and is of broad interest to the community

---

### Official Review · Reviewer_zKaR · 2023-11-22

**Novelty:** 6
**Technical Quality:** 5

**Review:**

The paper presents an evaluation of GDPR compliance and its impact on user privacy, with a focus on the Interactive Advertising Bureau Europe's TCF. This framework aims to provide a standard for sharing users' privacy consent choices among digital advertising market participants and is widely used across the internet. The study involves a thorough examination of 2230 websites using TCF and their compliance with GDPR regulations, particularly regarding the recording of user consent choices.

The study emploies robust methodologies to investigate a critical aspect of online privacy and data protection. It offers an analysis of GDPR compliance in the context of TCF, a relevant topic given the widespread use of TCF in digital advertising.

The paper is well-structured and clearly presents its findings, methodology, and the implications of its results. The data is meticulously analyzed and presented in an accessible format, making the complex subject matter understandable.

This study is likely unique in its focus on the TCF's compliance with GDPR. It fills a gap in existing research by not only assessing whether websites record user consent correctly but also exploring the nuances of 'legitimate interest' as a legal basis for data processing.

The findings are significant, especially for regulators, publishers, and privacy advocates. The study reveals that while most websites using TCF record user consent properly, a substantial number (72.8%) claim 'legitimate interest' to override user consent choices. This raises concerns about the effectiveness of TCF in protecting user privacy.

Pros

Examines a large sample of websites (2230) for a thorough analysis.

Addresses a critical and timely issue in digital privacy and data protection.

Highlights the widespread use of 'legitimate interest' and its implications for user consent.

Offers valuable insights for regulators and publishers to monitor compliance and detect non-compliance.

Cons

While the use of 'legitimate interest' is extensively discussed, the paper might benefit from a deeper exploration of its varied applications and user perspectives.

The study identifies issues with the IAB decoder, which could impact the accuracy of compliance assessments.

The focus is predominantly on European websites, which may limit its applicability in other jurisdictions with different privacy regulations.

**Questions:**

1. Could you elaborate on how 'legitimate interest' is often interpreted and applied by websites? Understanding the nuances of this legal basis could clarify whether its frequent use is a loophole in GDPR compliance or a legitimate practice.

2. The study mentions potential issues with the IAB TC String Decoder. How might these issues have affected your findings, and could resolving these issues change the study's conclusions regarding GDPR compliance?

3. Your study focuses on European websites. Do you believe your findings are representative of global trends in GDPR compliance, or are they specific to the European context?

4. Your research indicates that most users disagree with the application of 'legitimate interest' for data collection. Could you provide more details on how user perceptions were gauged and how these perceptions might influence the interpretation of your findings?

**Ethics Review Description:**

I've selected "No"

**Reviewer Confidence:**

3: The reviewer is confident but not certain that the evaluation is correct

**Scope:**

4: The work is relevant to the Web and to the track, and is of broad interest to the community

---

### Official Review · Reviewer_CQbu · 2023-11-23

**Novelty:** 6
**Technical Quality:** 6

**Review:**

The paper studies the Advertising Bureau Europe’s Transparency and Consent Framework (TCF), and in particular how websites implement the current version (2.1). The authors conclude that most websites comply with the framework rules, but many rely on the legitimate-interests legal basis and therefore may not actually be GDPR compliant.

I think the technical discussion is good, but some of the legal discussion feels confused. For example the paper talks about the 12 purposes (which is defined by the TCF) as if it is of equal standing to the legal bases in the GDPR. Whereas the TCF is not legally enforcable and is disputed, but the GDPR is law. It would be better to distinguish what the GDPR says from how the IAB interpret it. Similarly the paper stats that opt-in consent is required, but that's not the cases when one of the other legal bases is used.

The experiments appear to be well executed and the results are interesting; particularly that TCF v2.2 seems to forbid common behaviour. It should have significant impact on policy.

**Questions:**

- Of the claims made, which are clearly required by GDPR as compared to the IAB interpretation?

**Reviewer Confidence:**

3: The reviewer is confident but not certain that the evaluation is correct

**Scope:**

4: The work is relevant to the Web and to the track, and is of broad interest to the community

---

### Official Review · Reviewer_aDoZ · 2023-11-23

**Novelty:** 2
**Technical Quality:** 4

**Review:**

***Paper summary:***

This paper conducts an evaluation of website compliance with the Transparency and Consent Framework (TCF) and its impact on user privacy, covering 2,230 TCF-utilizing websites. Key findings indicate that 72.8% of TCF-compliant websites justify overriding user consent through the legitimate interest claim. This study is the first to explore TCF compliance post-versions 2.0 and 2.1, offering insights crucial for regulators and market participants navigating GDPR compliance and the IAB Tech Lab's Global Privacy Platform.



***Detailed comments for authors***

Thank you for submitting your work.
The paper undertakes a study to assess TCF compliance, revealing that only 2.2% of websites failed to adhere to users' consent choices, an improvement compared to prior works [40] that analyzed TCF compliance. The paper provides a comprehensive overview of the evolving TCF compliance landscape, demonstrating a positive trend in reduced user choice decline compared to previous studies.

The authors also delve into the use of legitimate interest, a concern raised in previous works. Notably, they explore its combination with specific purposes, highlighting potential violations on 16 websites out of the 2,230 studied (approximately 0.7% of the visited websites). While the paper effectively investigates the evolution of TCF compliance, I regret to note that the scientific contribution appears insufficient for me to recommend acceptance.

**Questions:**

-  How does your work compare to previous studies, and what are the novel contributions?

  - You analyzed both domains and pages within these domains, providing separate results for each.
When detecting a violation in a given domain, can there still be compliant pages within the same domain?

  -  In Line 635, where it states, "we determined that 2,315 crawls to 48 distinct domains qualified for this analysis," does this imply an average of 48 pages per domain?  are all visited pages belong to the same domain ?

  -  The paper's writing needs improvement,  some sections are challenging to follow.


   ** Minor:**
   -  Could you clarify how you derived the 1.3% (2.2%) value in Table 2?

**Ethics Review Description:**

I did not identify any ethical issues in the paper.

**Reviewer Confidence:**

3: The reviewer is confident but not certain that the evaluation is correct

**Scope:**

4: The work is relevant to the Web and to the track, and is of broad interest to the community

---

### Official Review · Reviewer_Lrgr · 2023-11-24

**Novelty:** 2
**Technical Quality:** 5

**Review:**

In this paper, the authors use automated crawls to opt-out of tracking by websites in the EU to assess whether these websites are storing compliant TCF strings. The authors conclude that most sites are correctly recording consent choices, but that websites abuse "legitimate interest" to ignore users' choices.

In general, it is important to study compliance with privacy laws, and this study fits within a recent body of work studying GDPR compliance in Europe. This work updates previous studies that found, arguably, more widespread non-compliance.

Unfortunately, the authors undercut the relevance of their own paper. The authors primary finding is that "legitimate interest" is being abused to facilitate ad tracking and targeting. However, as the authors note in 4.4, this is a known problem that has already been litigated, and TCF 2.2 will forbid this practice. In this light, the papers' findings seem to be too late.

The authors claim that their crawler stores HTTP requests and the cause that generated each request (e.g., the script that triggered it). However, unless I'm missing something, the authors don't utilize this data at all. This is a huge missed opportunity. Just because consent is stored in a cookie by the CMP does not mean third-parties are reading or receiving this information. Analyzing the parameters sent to third-parties, and why they were sent, would add significant depth to the paper.

Specific Comments:

1, "The high-level insight from our results is that most TCF sites are legally compliant." --- This is a bold proclamation, given that legal compliance stretches beyond asking for and recording consent. It also encompasses actually honoring consent information, e.g., not recording data or not serving targeted ads when consent is declined. It may be safer for the authors to stick to the facts rather than make broad proclamations, i.e., 97% of the sites in the sample asked for and recorded consent correctly.

2.2, "Given that each version of the TCF builds on the previous versions, our findings are still relevant." --- This is a true statement, but it also means prior work on TCF is still relevant as well. The authors claim that their study is the first to examine TCF 2.1 and 2.2, but its not clear what this contributes given that the authors are not actually investigating the specific changes that were implemented in TCF 2.1 and 2.2. Rather, this work seems like a straightforward replication of prior work that has already examined compliance with consent choices in the EU.

2.3: This section is really unconvincing. I'm very familiar with prior studies of GDPR compliance via TCF, and I'm not convinced this study is treading any new ground. Again, this study seems strongest when positioned as a replication: works have studied TCF consent signals in ~2020 and ~2021, but compliance practices may have changed since then.

4.1: DeepSee.io does not appear to be a public data source. How did the authors get access to it?

Table 3: A more informative way to perform this analysis is to plot the compliance rate vs. Tranco rank, rather than arbitrarily analyzing domains with rank > 5000.

**Questions:**

4.1: The description of the crawl process is somewhat ambiguous. Did the authors' crawler visit the homepage for each website? How were subpage links selected?

4.1: It is good practice to report the version number and date of the Tranco list used in studies.

4.1: When was the crawl conducted?

4.2: This section is a bit confusing. Is this section only examining consent strings in cookies, or is this also examining consent strings in HTTP parameters? Also, were there websites that stored in the consent string in multiple cookies, and if so, were the values consistent?

4.2: Is it possible that any of the cases where the TC string encoded "consent given" were the result of errors in the crawler, i.e., it clicked the consent button instead of the decline button? In other words, are the authors 100% sure that these are instances of non-compliance, or could crawler error be the cause? I am especially worried about this in the case of Ringier, since the non-complaince rate was 100%; did the crawler uniformly malfunction in some way when it encountered these banners?

**Reviewer Confidence:**

4: The reviewer is certain that the evaluation is correct and very familiar with the relevant literature

**Scope:**

4: The work is relevant to the Web and to the track, and is of broad interest to the community

---

### Decision · Program_Chairs · 2024-01-22

**Decision:**

Accept (Oral)

**Comment:**

Our decision is to accept. Please see the AC's review below and improve the work considering that and the reviewers' feedback for cemera-ready submission.

"This work analyzes compliance with TCF. The authors uncover interesting findings relative to the legitimate interest rationale. The reviewers maintain that the work's novelty is unclear but the work as being sound. After reading through the conversations, I agree with the assessments. The authors are encouraged to clarify the novelty of their work and address the methodological concerns raised. I recommend accept."